

# Bootstrapping mixed correlators in three-dimensional cubic theories II

Stefanos R. Kousvos[1,2] and Andreas Stergiou[3*]

**1** Department of Physics, University of Crete, Heraklion GR-70013, Greece
**2** Institute of Theoretical and Computational Physics (ITCP),
Department of Physics, University of Crete, 70013 Heraklion, Greece
**3** Theoretical Division, MS B285, Los Alamos National Laboratory,
Los Alamos, NM 87545, USA

★ andreas@lanl.gov

## Abstract

Conformal field theories (CFTs) with cubic global symmetry in 3D are relevant in a variety of condensed matter systems and have been studied extensively with the use of perturbative methods like the $\varepsilon$ expansion. In an earlier work, we used the nonperturbative numerical conformal bootstrap to provide evidence for the existence of a previously unknown 3D CFT with cubic symmetry, dubbed "Platonic CFT". In this work, we make further use of the numerical conformal bootstrap to perform a three-dimensional scan in the space of scaling dimensions of three low-lying operators. We find a three-dimensional isolated allowed region in parameter space, which includes both the 3D (decoupled) Ising model and the Platonic CFT. The essential assumptions on the spectrum of operators used to provide the isolated allowed region include the existence of a stress-energy tensor and the irrelevance of certain operators (in the renormalization group sense).



# 1  Introduction

Cubic scalar theories possess global symmetry described by the 48-element group $C_3 = \mathbb{Z}_2^3 \rtimes S_3 \simeq S_4 \times \mathbb{Z}_2 \subset O(3)$, where $S_n$ is the permutation group of $n$ objects. Their physical interest in 3D ($d = 3$ Euclidean dimensions) stems from their applications to finite temperature magnetic and structural phase transitions [1–5]. There exist a plethora of methods for studying them, e.g. the $\varepsilon$ expansion [5–11], the exact renormalization group [12], Monte Carlo simulations [13] and, more recently, the numerical conformal bootstrap [14–16]. Most studies until recently were performed with the $\varepsilon$ expansion, which provided up to six-loop resummed estimates for the experimentally observable critical exponents $\beta$ and $\nu$ [6]. Interestingly, the critical exponents of the $\varepsilon$ expansion, whose numerical value is almost degenerate with those of the $O(3)$ model, agree with experiments for magnetic phase transitions but are in tension with experiments for structural phase transitions [3, 17–20], which appear to be closer to Ising rather than $O(3)$ exponents.[1]

The study of scalar field theories with cubic symmetry via the numerical conformal bootstrap has a short history, although the bootstrap method is well-suited to the study of such theories.[2] The core constraints of any numerical bootstrap study are crossing symmetry, i.e. the statement of associativity of the operator product expansion (OPE), and unitarity.[3] Imposing these constraints, one may find bounds on allowed values of scaling dimensions of operators as well as coefficients in the OPE. In certain circumstances, these bounds are strong enough to confine scaling dimensions into isolated allowed regions in parameter space (islands), thus providing a calculation of said scaling dimensions which are directly linked to critical exponents. This strategy has been previously applied with great success to $O(N)$ symmetric CFTs. More specifically, in the case of $O(1)$ (i.e. the Ising model) it provided the most precise calculation of the critical exponents to date [27–31]. Extending this line of work to cubic-symmetric theories leads to the prediction of a new fixed point [15, 16], corresponding to the theory we will call "Platonic CFT". As explained in detail in [15, 16], this CFT may be relevant for cubic magnets and the structural phase transition of SrTiO$_3$. It may also be relevant for the phase transitions discussed in [32], whose critical exponents $\beta = 0.306(2)$, $\gamma = 1.185(13)$, $\delta = 4.857(30)$ for LEPMO and $\beta = 0.312(11)$, $\gamma = 1.177(17)$, $\delta = 4.776(70)$ for LYPMO agree very well with our determinations $\beta = 0.308(2)$, $\gamma = 1.167(8)$, $\delta = 4.792(11)$ for the Platonic CFT [16], although it should be noted that, according to [32], the crystals LEPMO and LYPMO they used crystallized in the orthorombic and not the cubic crystal system.

In our previous work we performed a mixed-correlator bootstrap analysis of four-point functions involving the leading scalar operators $\phi_i$ and $X_{ij}$ [16], where $\phi_i$ transforms as a vector and $X_{ij}$ as another nontrivial irreducible representation of the cubic group (see below). Our essential assumption was that the scaling dimension of $X$ saturated the bound that was obtained for that operator dimension using the single-correlator $\langle \phi\phi\phi\phi \rangle$ bootstrap in [15]. (We reproduce that bound in Fig. 1 for the reader's convenience.) The result of [16] was an island in the $\Delta_\phi$-$\Delta_S$ plane, where $S$ is the leading scalar singlet of the theory, obtained by further assuming that the next-to-leading scalar $S'$ and $X'$ operators are irrelevant (for $S'$ the assumption was $\Delta_{S'} \geqslant 3.7$).

In this work we first study the effect of assumptions on operators that are exchanged in the $\phi_i \times \phi_j$ OPE, where $\phi_i$ is the order parameter field which is a scalar under spatial rotations

---

[1]For extensive discussions on this matter see [15, 16] and references therein, and for a proposed resolution within the perturbative renormalization group framework see [21].

[2]We note that analytic bootstrap approaches in $d = 4 - \varepsilon$ dimensions, such as the Mellin bootstrap [22] and the large-spin bootstrap [23, 24], have produced critical exponents identical to those of the ordinary diagrammatic $\varepsilon$ expansion.

[3]See [25] for an introduction and [26] for a review.

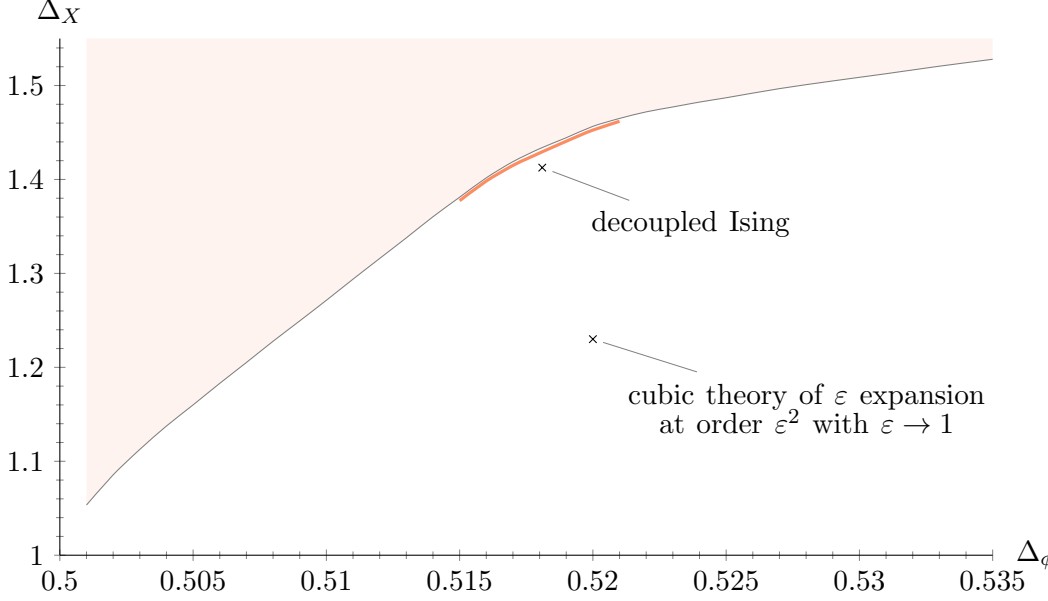

Figure 1: Upper bound on the dimension of the leading scalar $X$ operator in the $\phi_i \times \phi_j$ OPE. The red area is excluded. The gray bound is obtained using `PyCFTBoot` [33] with `mmax=6`, `nmax=9`, `kmax=36`, and `lmax=26`, while the for the red bound we use stronger numerics with `mmax=10`, `nmax=13`, `kmax=50`, and `lmax=40`. The decoupled Ising theory as well as the cubic theory of the $\varepsilon$ expansion are also indicated in the allowed region.

and transforms under the three-dimensional vector representation of the cubic group. In this part we still assume that $\Delta_X$ saturates the bound of Fig. 1. The $\phi_i \times \phi_j$ OPE can be written as

$$\phi_i \times \phi_j \sim \delta_{ij} S + X_{(ij)} + Y_{(ij)} + A_{[ij]}, \tag{1}$$

where $S$, $X_{(ij)}$ and $Y_{(ij)}$ are even-spin operators and $A_{[ij]}$ are odd-spin operators. If the cubic group is viewed as a subgroup of $O(3)$, $X$ and $Y$ form respectively the diagonal and non-diagonal parts of the two-index traceless-symmetric representation of $O(3)$ (conveniently thought of as a $3 \times 3$ matrix). Due to the fact that the cubic group is discrete, there is no global-symmetry conserved current in $A_{[ij]}$. The leading spin-two singlet in any local CFT is the stress-energy tensor, whose dimension is fixed to be equal to $d$ as a result of conservation. Below we will examine the effects of assuming that there is a stress-energy tensor $T_{\mu\nu}$ and a gap in the scaling dimension of $T_{\mu\nu}$ and the next-to-leading spin-two singlet, $T'_{\mu\nu}$.

The primary aim of this work is to eliminate the assumption that the leading scalar $X$ operator lies on the bound of Fig. 1. To that end, we study unitarity and crossing symmetry constraints on a system of correlators which consists of $\langle \phi\phi\phi\phi \rangle$, $\langle \phi\phi SS \rangle$ and $\langle SSSS \rangle$, where $S$ is the scalar singlet with the lowest scaling dimension (besides the obligatory unit operator) that appears in the OPE of $\phi_i$ with itself. Still assuming the existence of $T_{\mu\nu}$ and a gap to the dimension of $T'_{\mu\nu}$, along with further assumptions on some specific operators listed below, we are able to find a three-dimensional convex island that includes the decoupled Ising and the Platonic CFT. Within the set of assumptions we are using, we are unable to find two separate islands, one for each theory.

The structure of the present work is as follows. In section 2 we analyze the required group theory and obtain the constraints resulting from a single correlator system. In section 3 we analyze the system of multiple correlators involving $\phi$ and $S$, and derive numerically the three-dimensional island.

## 2 Single correlator

### 2.1 OPE, four-point function, and crossing equation

In order to work out the required tensor structures of the global symmetry for the four-point function, it is rather convenient to work with the cubic group as a subgroup of $O(3)$. Schematically, the OPE of an $O(3)$ vector with itself takes the form

$$\phi_i \times \phi_j \sim \delta_{ij} S + T_{(ij)} + A_{[ij]}, \tag{2}$$

where $S$ is the singlet, $T$ the traceless-symmetric and $A$ the antisymmetric representation. We note that in the cubic case the traceless-symmetric representation splits into its diagonal and non-diagonal parts; each one furnishes an irreducible representation (irrep) of the cubic group. Thus we have

$$\phi_i \times \phi_j \sim \delta_{ij} S + X_{(ij)} + Y_{(ij)} + A_{[ij]}, \tag{3}$$

where $X$ corresponds to the diagonal irrep and $Y$ to the non-diagonal irrep. The CFT of three decoupled Ising models has cubic symmetry. In that case the $X$ operators are given by sums of operators from each Ising model, while the $Y$ operators involve sums of products of operators from each Ising model. The lowest-dimension scalar $X$ and $Y$ operators in the decoupled Ising theory have scaling dimensions $\Delta_X = \Delta_\epsilon \approx 1.4126$ and $\Delta_Y = 2\Delta_\sigma \approx 1.0362$, respectively.

We wish to study the four-point function $\langle \phi_i \phi_j \phi_k \phi_l \rangle$. In order to decompose it onto irreps of the cubic group we use (3) and get, in the $12 \to 34$ channel,

$$\langle \phi_i \phi_j \phi_k \phi_l \rangle \sim \delta_{ij}\delta_{kl}\langle SS \rangle + \langle X_{ij}X_{kl} \rangle + \langle Y_{ij}Y_{kl} \rangle + \langle A_{ij}A_{kl} \rangle. \tag{4}$$

Noticing that that the tensor structures of the $X$ and $Y$ representations must add up to the one of the $T$ representation of $O(3)$, and demanding that they be orthogonal to each other as well as diagonal/non-diagonal, respectively, we obtain the following expressions for the global symmetry projectors of the four-point function:

$$P^S_{ijkl} = \tfrac{1}{3}\delta_{ij}\delta_{kl}, \qquad P^X_{ijkl} = \delta_{ijkl} - \tfrac{1}{3}\delta_{ij}\delta_{kl},$$
$$P^Y_{ijkl} = -\delta_{ijkl} + \tfrac{1}{2}(\delta_{ik}\delta_{jl} + \delta_{il}\delta_{jk}), \qquad P^A_{ijkl} = \tfrac{1}{2}(\delta_{il}\delta_{jk} - \delta_{ik}\delta_{jl}), \tag{5}$$

where the numeric prefactor of each projector is chosen in such a way that they are orthonormal to each other. It can easily be checked that $P^X_{ijkl} + P^Y_{ijkl} = P^T_{ijkl}$. With this in hand we have everything required to derive the crossing equations.

The crossing equation follows from demanding that the $12 \to 34$ decomposition of the four-point function is equal to the $14 \to 32$ one. We identify four equations, corresponding to terms multiplying $\delta_{ij}\delta_{kl}$, $\delta_{il}\delta_{jk}$, $\delta_{ik}\delta_{jl}$ and $\delta_{ijkl}$ which must independently be equal to zero. Hence, we arrive at our system of crossing equation sum rules:

$$\sum_{Y^+} \lambda^2_{O_Y} F^-_{\Delta,\ell} + \sum_{A^-} \lambda^2_{O_A} F^-_{\Delta,\ell} = 0,$$

$$\tfrac{1}{3}\sum_{S^+} \lambda^2_{O_S} F^-_{\Delta,\ell} - \tfrac{2}{3}\sum_{X^+} \lambda^2_{O_X} F^-_{\Delta,\ell} + \sum_{Y^+} \lambda^2_{O_Y} F^-_{\Delta,\ell} - \sum_{A^-} \lambda^2_{O_A} F^-_{\Delta,\ell} = 0,$$

$$\tfrac{1}{3}\sum_{S^+} \lambda^2_{O_S} F^+_{\Delta,\ell} - \tfrac{2}{3}\sum_{X^+} \lambda^2_{O_X} F^+_{\Delta,\ell} - \sum_{Y^+} \lambda^2_{O_Y} F^+_{\Delta,\ell} + \sum_{A^-} \lambda^2_{O_A} F^+_{\Delta,\ell} = 0,$$

$$\tfrac{1}{3}\sum_{S^+} \lambda^2_{O_S} F^-_{\Delta,\ell} + \tfrac{4}{3}\sum_{X^+} \lambda^2_{O_X} F^-_{\Delta,\ell} = 0, \tag{6}$$

where

$$F^\pm_{\Delta,\ell} = v^{\Delta_\phi} g_{\Delta,\ell}(u,v) \pm u^{\Delta_\phi} g_{\Delta,\ell}(v,u) \tag{7}$$

and $g$ is the corresponding conformal block for the exchange of an operator with scaling dimension $\Delta$ and spin $\ell$, while $u = \frac{x_{12}^2 x_{34}^2}{x_{13}^2 x_{24}^2}$ and $v = \frac{x_{14}^2 x_{23}^2}{x_{13}^2 x_{24}^2}$ are the usual conformal cross ratios and $x_{ij} = \sqrt{(x_i - x_j)^2}$. Lastly, the plus or minus superscript on operators being summed over indicates summation over even or odd spins, and for the OPE coefficients $\lambda_O$ is shorthand for $\lambda_{\phi\phi O}$.

## 2.2 Numerical results

Before proceeding, we note that a collection of results regarding bootstrap bounds obtained with assumptions on conserved currents for various symmetry groups have appeared previously in [34]. Our main assumption in this section is that the lowest-dimension scalar operator in the $X$ irrep saturates its corresponding bootstrap bound in Fig. 1.[4] The reason for this assumption is twofold. Firstly, bootstrap intuition tells us that kinks (abrupt changes in slope) in bootstrap bounds correspond to the positions in parameter space where physical CFTs live (which is precisely how the Ising and $O(N)$ CFTs were first discovered in the context of the conformal bootstrap). Such a kink is indeed observed in Fig. 1. Secondly, such an assumption is needed to obstruct the enhancement of the symmetry from cubic to $O(3)$. Such an enhancement requires $\Delta_X = \Delta_Y$, something that cannot happen because the bound on $\Delta_Y$ is much below the bound on $\Delta_X$ [15, Fig. 3].

The points used to saturate the $X$ sector bound are calculated with a vertical precision of $10^{-9}$. To produce the required xml files we used PyCFTBoot [33] with the following parameters: mmax=6, nmax=9, kmax=36, and lmax=26. All the plots presented in this section have a vertical precision of $10^{-6}$. For the optimization of the bootstrap problem we use SDPB [39,40].

First we obtain a plot, Fig. 2, using the following assumptions:

(S-1) saturation of $X$ bound of Fig. 1,
(S-2) existence of stress-energy tensor $T_{\mu\nu}$, i.e. $\Delta_{T_{\mu\nu}} = 3$,
(S-3) dimension of next-to-leading spin-two singlet, $T'_{\mu\nu}$, above 4, i.e. $\Delta_{T'_{\mu\nu}} \geqslant 4$,
(S-4) dimension of next-to-leading scalar singlet, $S'$, above 3, i.e. $\Delta_{S'} \geqslant 3$.

Note that the allowed region does not yet truncate on either side. Assumptions (S-3) and (S-4) are motivated by the extremal functional method [41] (for (S-4) see [15, Fig. 7]).

We may compare this to the allowed region studied in our previous mixed-correlator work (which did not assume the existence of a conserved stress-energy tensor); this is done in Fig. 3. Lastly, we may look at how the allowed region changes when also imposing a gap between the $X$ operator saturating the bootstrap bound of Fig. 1 and the next operator in the same irrep, $X'$—this is done in Fig. 4. There we see that the effect of raising the gap on $X'$ is that the allowed strip truncates on the left-hand side but remains unaffected on the right-hand side (compared to the strip in Fig. 2). We have checked that changing the gap on $S'$ between 3 and 3.7 leaves Fig. 4 essentially unchanged.

These gaps do not produce an island as we would ideally desire. In the next section we will see that, using a mixed-correlator system, we will be able to relax the assumptions on the $X$ irrep (i.e. we will no longer demand that it lives on the single correlator bound) and obtain an isolated allowed region in parameter space.

---

[4]Here we do not use the scalar singlet bound, for our theory of interest does not saturate that bound. It has been observed in practice that the bootstrap of subgroups of $O(N)$, for some value of $N$, always present scalar singlet sector bounds identical to $O(N)$ ones. Known examples include hypercubic and hypertetrahedral theories [15], MN and tetragonal theories [35] and $O(m) \times O(n)$ theories [36–38].

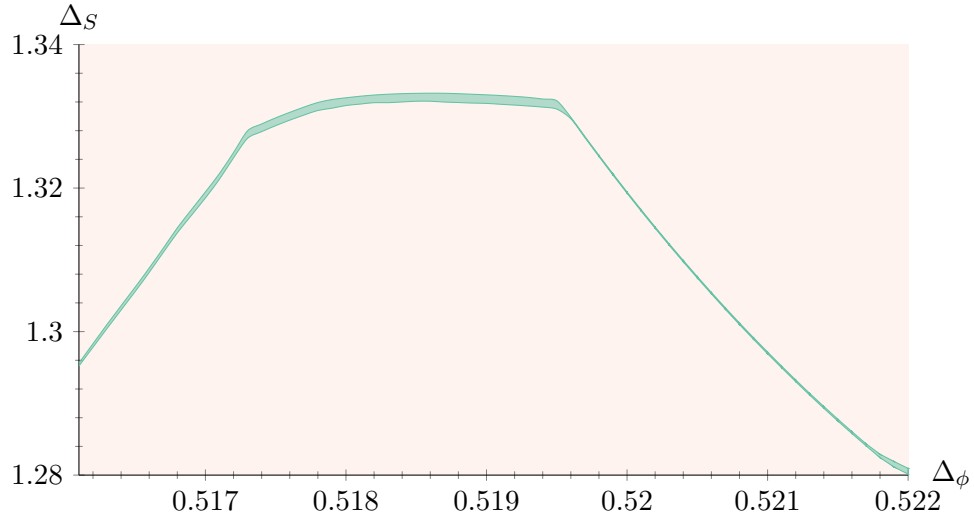

Figure 2: Plot of the allowed region (in green) with the assumptions (S-1)–(S-4). There appear to be two kinks in the allowed strip. It will become apparent from Fig. 3 that these two kinks arise at the points where this plot overlaps with the peninsula of allowed parameter space found in our previous work [16].

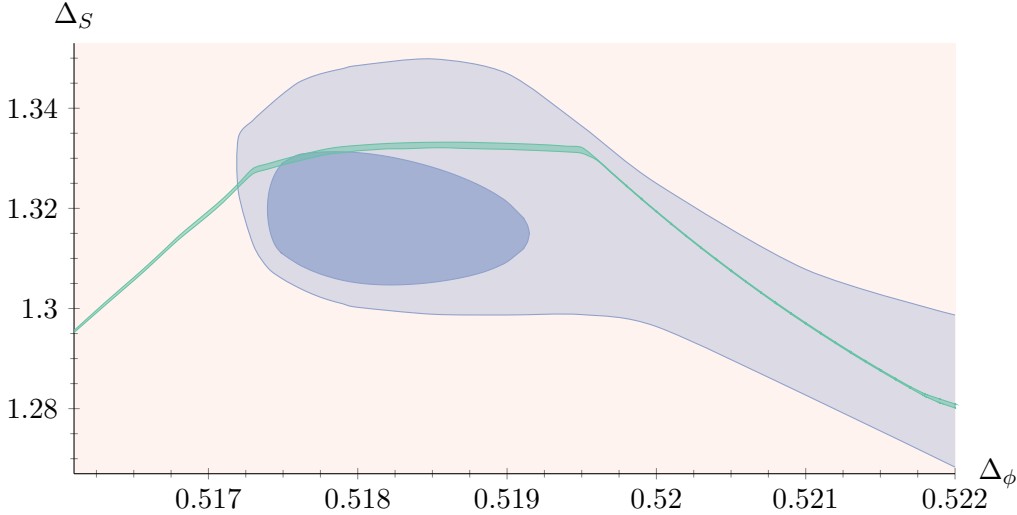

Figure 3: In this figure we observe the overlap between the single-correlator allowed region from this work (in green), and the allowed peninsula and island from our previous mixed-correlator work [16] (in blue). Both the peninsula and island assume that the theory lives on the $X$ sector single-correlator bound of Fig. 1, and also that the next-to-leading operator in the scalar $X$ sector has a scaling dimension of 3 or higher. Additionally, the island assumes that $S'$ has a dimension of 3.7 or higher. The peninsula assumes that $S'$ has a dimension of 3 or higher.

## 3 Multiple correlators

### 3.1 OPEs, four-point functions, and crossing equations

In this section we study the additional correlators needed for the three dimensional scan of parameter space. First, we must note that in this work we consider a different system of mixed correlators compared to our previous work [16]. This brings considerable simplifications in

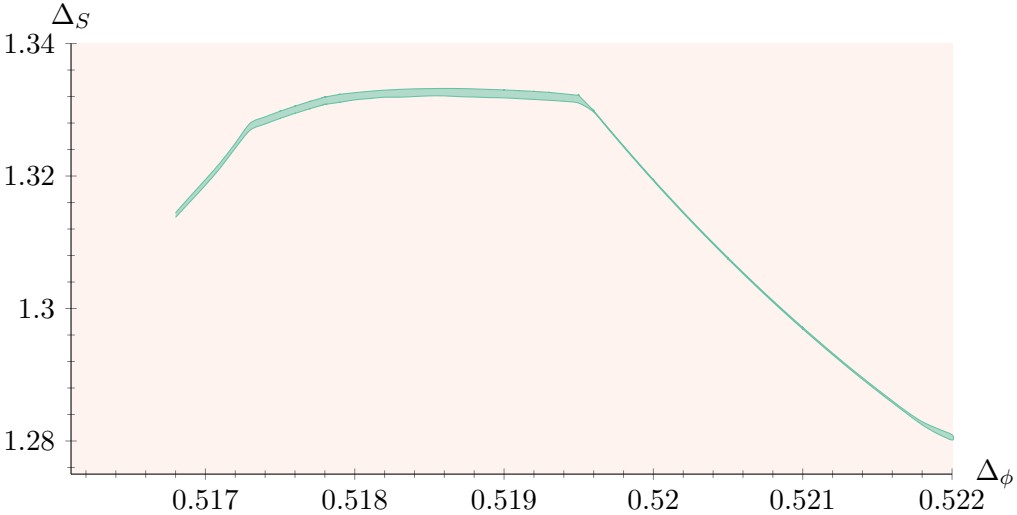

Figure 4: In this plot we assume, in addition to (S-1)–(S-4), that $X'$ has a dimension of 2.8 or higher. This extra assumption leads to the truncation of the allowed region on the left.

terms of the group theory structures required. We need the OPE of the vector operator with the scalar singlet $S$, and the OPE of the scalar singlet $S$ with itself. These follow trivially:

$$\phi_i \times S \sim \phi_i \,, \tag{8}$$

and

$$S \times S \sim S \,. \tag{9}$$

Using (3), (8) and (9) we obtain

$$\langle \phi_i S \phi_j S \rangle \sim \langle \phi_i \phi_j \rangle \,, \qquad \langle \phi_i \phi_j SS \rangle \sim \delta_{ij} \langle SS \rangle \,, \tag{10}$$

where we take the OPE of the first two and last two operators in the four-point function.

The crossing equations are

$$\sum_{S^+} \lambda^2_{SSO_S} F^{-SS,SS}_{\Delta,\ell} = 0 \,,$$

$$\sum_{V^\pm} \lambda^2_{\phi SO_V} F^{-\phi S,\phi S}_{\Delta,\ell} = 0 \,, \tag{11}$$

$$\sum_{S^+} \lambda_{\phi\phi O} \lambda_{SSO_S} F^{\mp \phi\phi,SS}_{\Delta,\ell} \pm \sum_{V^\pm} (-1)^\ell \lambda^2_{\phi SO_V} F^{\mp S\phi,\phi S}_{\Delta,\ell} = 0 \,,$$

where again the plus or minus superscripts on operators are used to indicate summation over even or odd spin operators, respectively, and

$$F^{\pm ij,kl}_{\Delta,\ell} = v^{\frac{\Delta_j + \Delta_k}{2}} g^{ij,kl}_{\Delta,\ell}(u,v) \pm u^{\frac{\Delta_j + \Delta_k}{2}} g^{ij,kl}_{\Delta,\ell}(v,u) \,. \tag{12}$$

The numerical bootstrap problem can now be set up along the lines described in [30].

## 3.2 Numerical results

Equipped with the machinery of mixed correlators, we may now obtain a closed isolated region in parameter space. This works as follows. It is important, just as in the previous section, to

obstruct the enhancement of the symmetry to $O(3)$. This is done by assuming that the scaling dimension of the first $X$ operator is equal to, or greater than 1.4126. This is in the allowed region of Fig. 1 for $\Delta_\phi \gtrsim 0.5165$, and it obstructs the enhancement of the symmetry to $O(3)$ due to the fact that $\Delta_X \geqslant 1.4126$ lies in the disallowed region for the scaling dimension of the leading scalar $Y$ sector operator for $\Delta_\phi$ in our region of interest (see [15, Fig. 3]).[5] Note that in [15] it was observed that all $X$ sector hypercubic kinks lie above the Ising model, which they approach from above as $N \to \infty$. (This also motivates our choice of the Ising gap in assumption (M-1) below.)

The assumptions we make are summarized below:

(M-1)  $\Delta_X \geqslant 1.4126$,
(M-2)  existence of conserved stress-energy tensor, $T_{\mu\nu}$, i.e. $\Delta_{T_{\mu\nu}} = 3$,
(M-3)  $\Delta_{T'_{\mu\nu}} \geqslant 4$,
(M-4)  $\Delta_{S'} \geqslant 3.7$,
(M-5)  $\Delta_{Y'} \geqslant 3.0$,
(M-6)  $\Delta_{\phi'} \geqslant 1.5$.

We also impose the equality of OPE coefficients $\lambda_{\phi\phi S} = \lambda_{\phi S\phi}$. Here $Y'$ is the next-to-leading scalar operator in the $Y$ irrep of the $\phi \times \phi$ OPE and $\phi'$ is the next-to-leading operator in the $\phi \times S$ OPE ($\phi$ is the leading one). The assumption (M-6) is the statement that the cubic in $\phi$ operator has dimension above the free theory dimension of $\phi^2 \phi_i$. With these assumptions we perform multiple scans: for each scan we fix a dimension $\Delta_Y$ for the leading $Y$ operator and find the lowest and highest admissible value for $\Delta_S$. This procedure leads to Fig. 5, which is obtained by finding the allowed region in the three-dimensional space parametrized by $(\Delta_\phi, \Delta_S, \Delta_Y)$ and consequently projecting it onto the $\Delta_Y$-$\Delta_S$ plane. Indicatively, we show three different slices of this three-dimensional space, obtained by fixing $\Delta_Y = 1$, $\Delta_Y = 0.98575$ and $\Delta_Y = 1.025$ and looking at the $\Delta_\phi$-$\Delta_S$ plane in Figs. 6, 7 and 8, respectively. The parameters we have used in PyCFTBoot [33] are, mmax=5, nmax=7, kmax=36, and lmax=26 . Depending on the position in Fig. 5, the vertical precision starts from 0.002 and goes up to 0.0001 (the increased precision is used at the edges of the island for increased smoothness).

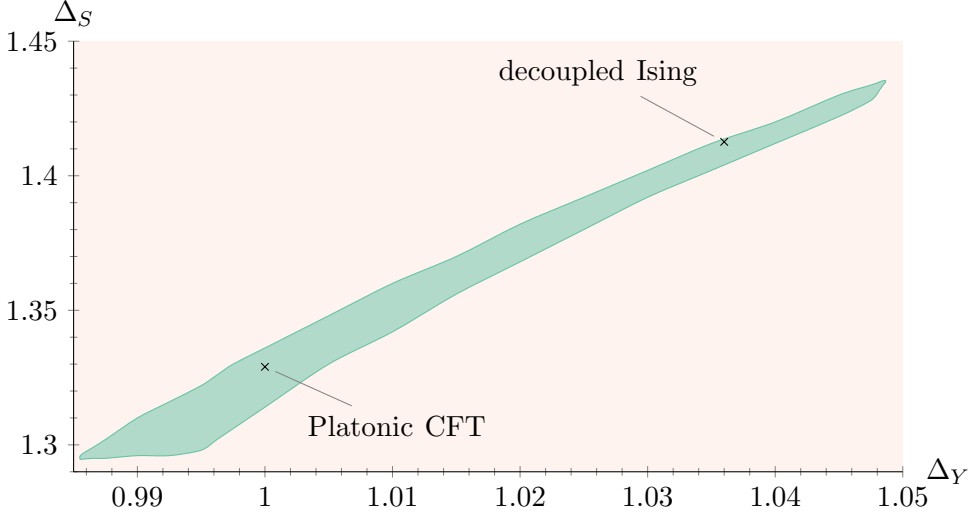

Figure 5:  Plot of the projection of the three-dimensional allowed region in $(\Delta_\phi, \Delta_S, \Delta_Y)$ onto the $\Delta_S$-$\Delta_Y$ plane using the assumptions (M-1)–(M-6).

---

[5]In the $O(3)$ theory $X$ and $Y$ have the same scaling dimension, for they combine to form the two-index symmetric traceless of $O(3)$.

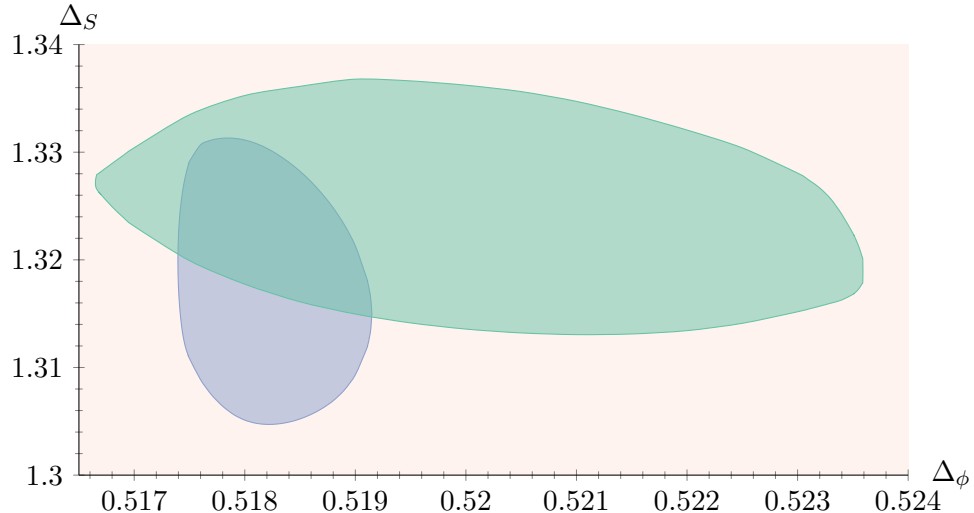

Figure 6: Plot of the allowed region in the $\Delta_\phi$-$\Delta_S$ plane, derived with $\Delta_Y = 1$ and using the assumptions (M-1)–(M-6). The blue island is that of [16, Fig. 7] with $\Delta_{S'} \geqslant 3.7$.

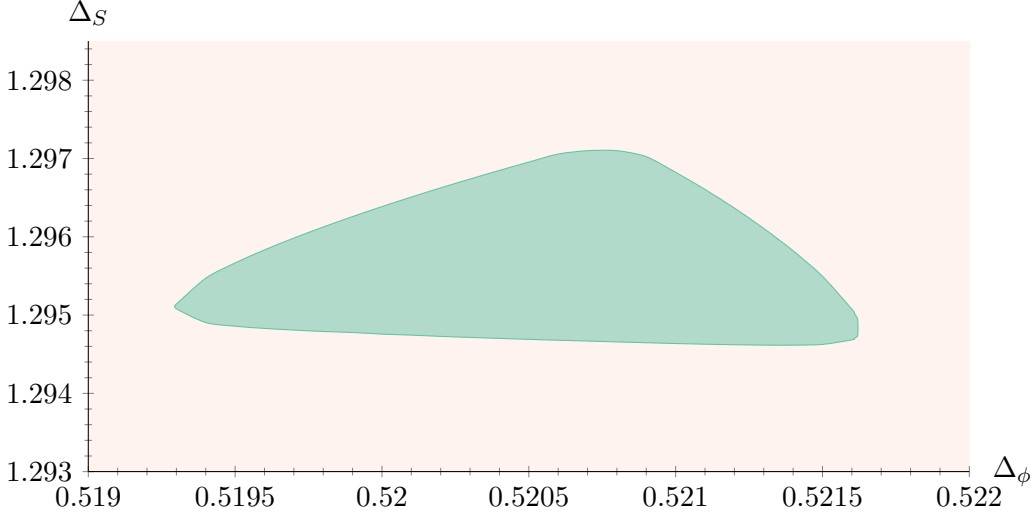

Figure 7: Plot of the allowed region in the $\Delta_\phi$-$\Delta_S$ plane, derived with $\Delta_Y = 0.98575$ and using the assumptions (M-1)–(M-6) .

We find that the right-hand side of the island in Fig. 5 truncates close to the Ising fixed point, whereas the left-hand side of the island truncates close to the Platonic fixed point. Note that, using the extremal functional method [41] on the $X$ sector single correlator bound of Fig. 1, we find a scaling dimension $\Delta_Y$ numerically very close to 1 for the Platonic CFT.[6] This is consistent with the island presented here. For $\Delta_Y = 1$ one may view Fig. 6, where the large green island corresponds to the one found in this work and the smaller blue one to that of [16, Fig. 7] with $\Delta_{S'} \geqslant 3.7$. The Platonic CFT must lie in the overlap of the two islands of Fig. 6.

As can be seen from Fig. 7, the island shrinks in size as we approach the left-most edge of the allowed region of Fig. 5, but without approaching $\Delta_\phi = 0.5165$ (the $\Delta_\phi$ above which

---

[6]We note that although both decoupled Ising and Platonic CFT are denoted by a cross in Fig. 5, the position of the Platonic CFT is approximate and not known with the precision of the decoupled Ising CFT.

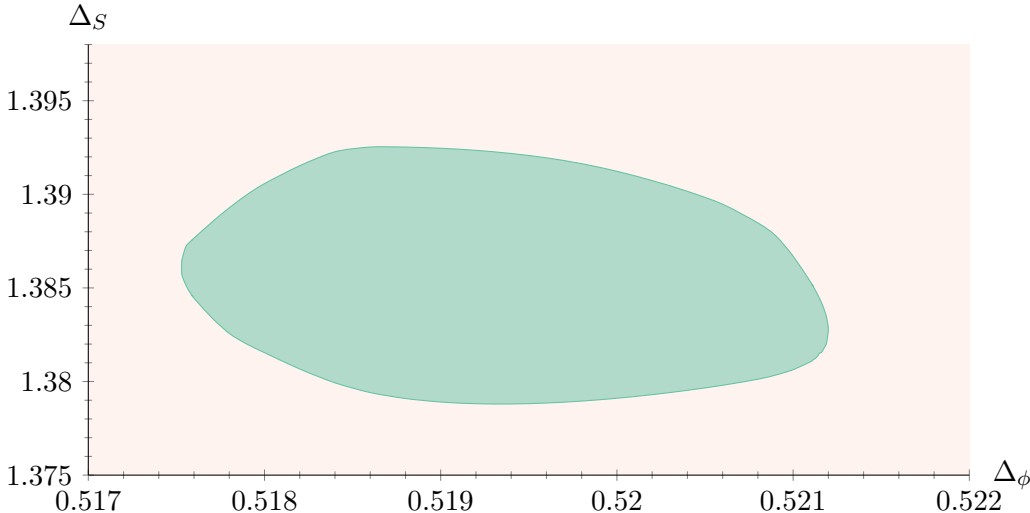

Figure 8: Plot of the allowed region in the $\Delta_\phi$-$\Delta_S$ plane, derived with $\Delta_Y = 1.025$ and using the assumptions (M-1)–(M-6).

(M-1) becomes valid). Similarly, Fig. 8 shows that the island shrinks in size as we approach the right-most edge of the allowed region of Fig. 5, again without approaching $\Delta_\phi = 0.5165$. Hence, our three-dimensional island is isolated because of assumptions (M-2)–(M-6), and assumption (M-1) simply serves the purpose of precluding the enhancement of the global symmetry to $O(3)$.

## 4 Concluding remarks

In this work we studied the parameter space surrounding the Platonic CFT. This was done using a mixed-correlator bootstrap, with the external operators being scalar (spin-zero) operators that transform in the vector and singlet representation of the cubic group. In our previous work we had studied a mixed-correlator system involving the vector of the cubic group, but instead of the singlet we had considered a scalar in the diagonal representation we call $X$ [16]. That work had produced an island in the $\Delta_\phi$-$\Delta_S$ plane, but it involved the assumption that $\Delta_X$ saturated its bootstrap bound (shown here in Fig. 1). The main motivation of this work was to remove that assumption. Even without it, we find a three-dimensional island, two-dimensional projections of which can be seen in Figs. 5 and 6. Our results further establish and solidify the existence of the Platonic CFT.

The island we find in this work includes the Platonic and the decoupled Ising CFT (see Fig. 5).[7] We were not able to separate the allowed region into two islands, one around each CFT. This demonstrates that the low-lying spectrum of the two theories is very similar, at least in the sectors we have examined, although their critical exponents are manifestly different. The crucial spectrum assumption that can be used to distinguish these two CFTs and allow us to obtain two distinct islands around them remains unknown.

An important result established here is that the Platonic CFT contains a conserved stress-energy tensor, something that implies its locality. This result is nontrivial due to the fact that currently we do not have a microscopic understanding of the Platonic CFT, i.e. we are not aware of a Lagrangian that flows to it, as is the case, for example, for the Ising model in $d = 3$.

This work offers independent constraints on operators that have been already constrained

---

[7]We remind the reader that they both have the same global symmetry, namely cubic.

using other channels in our earlier work. More specifically, scalar singlets appear in the $\phi \times \phi$ and $X \times X$ OPEs, see [15] and [16], respectively, while they also appear in the $S \times S$ OPE we considered in this work. The overall consistency of our results is corroborated by our results here, which suggest that the dimension of the next-to-leading scalar singlet is well above marginality, specifically $\Delta_{S'} \approx 3.7$. This means that the Platonic CFT has only one relevant scalar singlet operator, and thus it corresponds to a critical theory in the usual classification.

This work also further supports the conclusion that the Platonic CFT is not the cubic theory found with the $\varepsilon$ expansion in $d = 4 - \varepsilon$. Indeed, the critical exponents determined for the cubic theory of the $\varepsilon$ expansion [6] are very different from those determined from our results in this work (using the overlap in Fig. 6 for example), which are consistent with our earlier results in [16].

An interesting future direction would be to perform a mixed-correlator bootstrap with $\phi$, $X$ and $S$ as external operators, and identify the assumptions with which we may obtain a three-dimensional island in the $(\Delta_\phi, \Delta_X, \Delta_S)$ space. It would also be of great interest to use the bootstrap to study the predictions of the $\varepsilon$ expansion for the cubic theory, i.e. use spectrum assumptions based on $\varepsilon$ expansion results in order to move into the allowed region of Fig. 1 and nonperturbatively analyze the cubic theory predicted by the $\varepsilon$ expansion.

# Acknowledgments

SRK would like to thank Slava Rychkov for his hospitality and support during his extended stay at the Institut des Hautes Études Scientifiques (IHES) where part of this work was completed; this stay was supported by the Simons Foundation grant 488655 (Simons Collaboration on the Nonperturbative Bootstrap) and by Mitsubishi Heavy Industries via an ENS-MHI Chair. SRK would also like to thank the IHES for their hospitality during his extended stay there. The research work of SRK is supported by the Hellenic Foundation for Research and Innovation (HFRI) under the HFRI PhD Fellowship grant (Fellowship Number: 1026). Research presented in this article was supported by the Laboratory Directed Research and Development program of Los Alamos National Laboratory under project number 20180709PRD1. The numerical computations in this paper were run on the LXPLUS cluster at CERN and the Metropolis cluster at the Crete Center for Quantum Complexity and Nanotechnology.

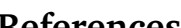

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
