# Peer review of "Bootstrapping Mixed Correlators in Three-Dimensional Cubic Theories II"

_SciPost Physics, doi:SciPost Phys. 8, 085 (2020)_

## Round 1 · Referee Report · Anonymous (Referee 1) · 2020-3-28

Strengths

  1. the plots produced in this paper come from the numerical bootstrap, so they are rigorous.

  2. the paper attempts to address a non-supersymmetric CFT in 3d, which is interesting because such theories are more likely to be relevant to nature (than higher d CFTs or those with supersymmetry).

  3. The paper focuses on a putative new CFT, that would be discovered using the bootstrap.

Weaknesses

  1. The entire goal of this paper is to try to study a theory called the Platonic CFT. The only evidence for this existence is a (rather dubious) kink, and the fact that with many extreme assumptions an island forms in the vicinity of that kink. However, there is no other motivation presented for this theory from either theory or experiment.

  2. One should keep in mind that if you impose enough gaps in a bootstrap study of any symmetry, you will always start finding kinks and islands. There is no reason to think that these correspond to anything physical though. The whole motivation of the bootstrap is that it should be rigorous, and thus as free of assumptions as possible. But imposing tons of unjustified gaps take away from that advantage.

  3. On a technical note, now that the bootstrap can be easily set up for any non-supersymmetric CFT using the autoBoot software, the motivation for deriving non-supersymmetric crossing equations (which would have automatically motivated previous studies, regardless of the numerical results) is much diminished. As such, future studies are only really motivated if the numerical results are particularly striking.

  4. The numerics have been run with relatively low precision by modern standards (nmax=9), so its difficult to know how converged the numerics are. With the new version of SDPB, much higher nmax should be possible.

Report

This paper attempts to find additional evidence for a putative 3d CFT with cubic symmetry, which is different from the standard cubic symmetry CFT that one studies with the epsilons expansion (and which is believed to be very close to the O(3) model in the RG sense, and also may describe real condensed matter systems). The previous evidence for this CFT was a kink observed in a plot by the same authors in a previous paper after various assumptions were imposed, that further became an island after more assumptions were imposed. In this paper, after imposing even more assumptions on the spectrum, they can find an island in a similar part of theory space without one of the previous assumptions (that the theory lived on the boundary of a previously computed numerical plot).

Requested changes

  1. It would be good to show some plots showing how converged the numerics are, e.g. by comparing plots with different values of nmax.

  • validity: good
  • significance: low
  • originality: low
  • clarity: high
  • formatting: good
  • grammar: good

Author:  Andreas Stergiou  on 2020-03-30  [id 782]

(in reply to Report 1 on 2020-03-28)

We would like to thank the referee for his/her comments on our manuscript. To address the referee's indicated weakness 1, we added some comments on the experimental motivation for the study of the Platonic CFT in the second paragraph of the introduction. This was thoroughly explained in our earlier work, to which the reader is referred for further reading.

The referee's weakness 2, that "if you impose enough gaps in a bootstrap study of any symmetry, you will always start finding kinks and islands" is wrong or at least misleading. In theories with operators transforming under a wide variety of irreducible representations of some global symmetry group, it is inevitable that to isolate special/physical theories one needs to make assumptions in many of these sectors. Even in the Ising model, assumptions of irrelevance need to be made in both Z_2-even and Z_2-odd scalar operators to obtain a bootstrap island.

The referee's weakness 3 is addressed by the potential relevance of the Platonic CFT to phase transitions that have been observed in experiments.

To address the referee's weakness 4, and also his/her requested change 1, we have modified Fig. 1 by adding an upper bound obtained with much stronger numerics. This shows that our bound is highly converged.

Attachment:

bootstrapping_mixed_cubic_correlators_II.pdf

---

## Round 1 · Referee Report · Anonymous (Referee 2) · 2020-3-30

Report

The authors consider a $3D$ CFT with cubic symmetry. The study of this theory is well motivated from experimental and theoretical points of view. In fact experiments for structural phase transitions suggest the existence of a non trivial fixed point with cubic symmetry, whose critical exponents are incompatible with the estimates obtained by epsilon expansion. The aim of the paper is to provide new evidences for the existence of this cubic theory, dubbed "Platonic CFT".

Evidences for the existence of the Platonic CFT were provided in a series of precedent numerical conformal bootstrap results. By imposing crossing symmetry of a four-point function, unitarity and some extra assumptions, bounds and island on some operator dimensions were obtained. However all previous results were based on one non-rigorous assumption: that the conformal dimension of one operator ($X_{ij}$) was fixed in terms of the dimension of another one ($\phi_i$). Following the bootstrap lore that existing theories typically saturate bootstrap bounds, the function $\Delta_{X}(\Delta_{\phi})$ was defined in order to saturate the bootstrap bound for the dimensions of the operator $X$ using a single correlator of $\langle \phi \phi \phi \phi \rangle$. However, strictly speaking, the true value of $\Delta_{X}$ is unlikely to exactly lay on top of this curve. Therefore this assumption gave rise to an error which was hard to quantify.

In this paper the authors managed to drop this assumption by trading it for more rigorous ones, like the existence of a stress tensor. They first show how the existence of the stress tensor, even in a single correlator setup, gives rise to very strong bounds. They further use this constraint (and few extra assumptions on gaps of other operators, which may be satisfied by the Platonic CFT) in a mixed-correlator setup and manage to obtain an allowed island —compatible with the previous bootstrap studies— in the space spanned by the dimensions of the three operators $\phi, X, S$. This is an important result since it provides a very strong evidence for the existence of the Platonic CFT. The paper is well written and the results are important and new. I am therefore happy to recommend the paper for publication. 

I have few suggestions.

1) In page 2, the operators $\phi$, $X$ and $S$ are introduced without definition. It would be nice either to say "as defined below", or to move the paragraph of formula (1.1) above.

2) Figure 4 would be quicker to read if it had a piece of the disallowed region in left of the plot. In other words I suggest to keep the same origin as in figure 2 and 3.

3) In section 2.2, I suggest to add a comment on the assumption of the gap above the stress-tensor. Is there a reason for this choice? What happens when different gaps are considered? Is the gap motivated by the extremal functional method?

4) Similarly to comment 3), in section 3.1, it would be interesting to know more about the choice of the gaps, namely assumptions (M-3), (M-4), (M-6).

5) The explanation “due to the fact that $\Delta_X \geq 1.4126$ lies in the disallowed region for scaling dimensions of $Y$ sector operators for $\Delta_\phi$ rather large (certainly well above our region of interest [10, Fig. 3]).” is difficult to understand. One may add that for the $O(3)$ model $X$ and $Y$ are the same operator. Also “for $\Delta_\phi$ rather large’’ can be interpreted as “only when $\Delta_\phi$ is rather large’’, while it should mean “for any $\Delta_\phi$ smaller than some rather large value’’.

  • validity: -
  • significance: -
  • originality: -
  • clarity: -
  • formatting: -
  • grammar: -

Author:  Andreas Stergiou  on 2020-05-22  [id 838]

(in reply to Report 2 on 2020-03-30)

We would like to thank this referee for their thorough and thoughtful report. We found the referee's suggestions very helpful and we modified our draft accordingly.

---

## Round 2 · List of Changes

Changes as outlined in responses to referee reports.

---

## Editorial Decision

published